# Consistent Aggregation of Objectives with Diverse Time Preferences Requires Non-Markovian Rewards

**Silviu Pitis**
University of Toronto and Vector Institute
`spitis@cs.toronto.edu`

## Abstract

As the capabilities of artificial agents improve, they are being increasingly deployed to service multiple diverse objectives and stakeholders. However, the composition of these objectives is often performed ad hoc, with no clear justification. This paper takes a normative approach to multi-objective agency: from a set of intuitively appealing axioms, it is shown that Markovian aggregation of Markovian reward functions is not possible when the time preference (discount factor) for each objective may vary. It follows that optimal multi-objective agents must admit rewards that are non-Markovian with respect to the individual objectives. To this end, a practical non-Markovian aggregation scheme is proposed, which overcomes the impossibility with only one additional parameter for each objective. This work offers new insights into sequential, multi-objective agency and intertemporal choice, and has practical implications for the design of AI systems deployed to serve multiple generations of principals with varying time preference.

## 1 Introduction

The idea that we can associate human preferences with scalar utility values traces back hundreds of years and has found usage in numerous applications [9, 71, 28, 49]. One of the most recent, and perhaps most important, is the design of artificial agents. In the field of reinforcement learning (RL), this idea shows up as the *reward hypothesis* [74, 67, 10], which lets us define objectives in terms of a discounted sum of Markovian rewards. While foundational results from decision theory [81, 62] and inverse RL [52, 54] justify the reward hypothesis when a single objective or principal is considered, complexities arise in multi-objective scenarios [61, 77]. The literature on social choice is largely defined by impossibilities [5], and multi-objective composition in the RL and machine learning literature is typically restrictive [68, 51], applied without clear justification [29], or based on subjective evaluations of empirical efficacy [21]. Addressing these limitations is crucial for the development of artificial agents capable of effectively serving the needs of diverse stakeholders.

This paper extends previous normative work in RL by adopting an axiomatic approach to the aggregation of objectives. The approach is based on a set of intuitively appealing axioms: the von Neumann-Morgenstern (VNM) axioms, which provide a foundation for rational choice under uncertainty; Pareto indifference, which efficiently incorporates individual preferences; and dynamic consistency, which ensures time-consistent decision-making. From these axioms, an impossibility is derived, leading to the conclusion that optimal multi-objective agents with diverse time preferences must have rewards that are non-Markovian with respect to the individual objectives. To address this challenge, a practical state space expansion is proposed, which allows for the Markovian aggregation of objectives requiring only one parameter per objective. The results prompt an interesting discussion on dynamic preferences and intertemporal choice, leading to a novel "historical discounting" strategy that trades off dynamic consistency for intergenerational fairness. Finally, it is shown how both our results can be extended (albeit non-normatively) to stochastic policies.

37th Conference on Neural Information Processing Systems (NeurIPS 2023).

The remainder of this paper is organized as follows: Section 2 motivates the problem by modeling human procrastination behavior as an aggregation of two objectives, work and play, and showing how a plan that appears optimal today may lead to the worst possible future outcome. Section 3 presents the axiomatic background and the key impossibility result. Section 4 presents the corresponding possibility result and a practical state expansion to implement it. Section 5 relates the results to intertemporal choice, proposes $N$-step commitment and historical discounting strategies for managing intergenerational tradeoffs, extends the results to stochastic policies, and discusses related topics in RL. Section 6 concludes with some final thoughts and potential future research directions.

## 2    Motivation: The Procrastinator's Peril

We begin with a numerical example of how the naive aggregation of otherwise rational preferences can lead to undesirable behavior. The example, which will be referred to throughout as the "Procastinator's Peril", involves repeated procrastination, a phenomenon to which the reader might relate. An agent aggregates two competing objectives: work and play. At each time step the agent can choose to either work or play. The pleasure of play is mostly from today, and the agent doesn't value future play nearly as much as present play. On the other hand, the consequences of work are delayed, so that work tomorrow is valued approximately as much as work today.

Let us model the agent's preferences for work and play as two separate Markov Decision Processes (MDP), each with state space $\mathcal{S} = \emptyset$ and action space $\mathcal{A} = \{\mathtt{w}, \mathtt{p}\}$. In the play MDP, we have rewards $R(\mathtt{p}) = 0.5$, $R(\mathtt{w}) = 0$ and a discount factor of $\gamma_{\mathtt{play}} = 0.5$. In the work MDP, we have rewards $R(\mathtt{p}) = 0$, $R(\mathtt{w}) = 0.3$ and a discount factor of $\gamma_{\mathtt{work}} = 0.9$. One way to combine the preferences for work and play is to value each trajectory under both MDPs and then add up the values. Not only does this method of aggregation seem reasonable, but it is actually *implied* by some mild and appealing assumptions about preferences (Axioms 1 and 3 in the sequel). Using this approach, the agent assigns values to trajectories as follows:

| | | | |
|---|---|---|---|
| $\tau_1$ | $\mathtt{p}, \mathtt{p}, \mathtt{p}, \mathtt{p}\ldots$ | $V(\tau_1) = \sum_t (0.5)^t \cdot 0.5$ | $= 1.00$ |
| $\tau_2$ | $\mathtt{w}, \mathtt{w}, \mathtt{w}, \mathtt{w}\ldots$ | $V(\tau_2) = \sum_t (0.9)^t \cdot 0.3$ | $= 3.00$ |
| $\tau_3$ | $\mathtt{p}, \mathtt{w}, \mathtt{w}, \mathtt{w}\ldots$ | $V(\tau_3) = 0.5 + 0.9 \cdot V(\tau_2)$ | $= 3.20$ |
| $\tau_4$ | $\mathtt{p}, \mathtt{p}, \mathtt{w}, \mathtt{w}\ldots$ | $V(\tau_3) = 0.75 + 0.9^2 \cdot V(\tau_2)$ | $= 3.18$ |

We see that the agent most prefers $\tau_3$: one period (and one period only!) of procrastination is optimal. Thus, the agent procrastinates and chooses to play today, planning to work from tomorrow onward. Come tomorrow, however, the agent is faced with the same choice, and once again puts off work in favor of play. The process repeats and the agent ends up with the least preferred alternative $\tau_1$.

This plainly irrational behavior illustrates the impossibility theorem. Observe that the optimal policy $\tau_3$ is non-Markovian—it must remember that the agent has previously chosen play in order to work forever. But any MDP has a stationary optimal policy [55], so it follows that we need rewards that are non-Markovian with respect to the original state-action space. Alternatively, we will see in Subsection 4.2 that we can expand the state space to make the optimal policy Markovian.

## 3    Impossibility of Dynamically Consistent, Pareto Indifferent Aggregation

**Notation**   We assume familiarity with Markov Decision Processes (MDPs) [55] and reinforcement learning (RL) [74]. We denote an MDP by $\mathcal{M} = \langle \mathcal{S}, \mathcal{A}, T, R, \gamma \rangle$, where $\gamma : \mathcal{S} \times \mathcal{A} \to \mathbb{R}^+$ is a state-action dependent discount function. This generalizes the usual "fixed" $\gamma \in \mathbb{R}$ and covers both the episodic and continuing settings [85]. We use lowercase letters for generic instances, e.g. $s \in \mathcal{S}$, and denote distributions using a tilde, e.g. $\tilde{s}$. In contrast to standard notation we write both state- and state-action value functions using a unified notation that emphasizes the dependence of each on the future policy: we write $V(s, \pi)$ and $V(s, a\pi)$ instead of $V^\pi(s)$ and $Q^\pi(s, a)$. We extend $V$ to operate on probability distributions of states, $V(\tilde{s}, \Pi) = \mathbb{E}_{s \sim \tilde{s}} V(s, \Pi)$, and we allow for non-stationary, history dependent policies (denoted by uppercase $\Pi, \Omega$). With this notation, we can understand $V$ as an expected utility function defined over prospects of the form $(\tilde{s}, \Pi)$. We use the letter $h$ to denote histories (trajectories of states and actions)—these may terminate on either a state or action, as may be inferred from the context. For convenience, we sometimes directly concatenate

histories, states, actions and/or policies (e.g., $hs$, $sa$, $s\Pi$, $a\Pi$) to represent trajectory segments and/or the associated stochastic processes. For simplicity, we assume finite $|\mathcal{S}|, |\mathcal{A}|$.

## 3.1 Representing rational preferences

This paper is concerned with the representation of aggregated preferences, where both the aggregation and its individual components satisfy certain axioms of rationality. We define the objects of preference to be the stochastic processes ("prospects") generated by following (potentially non-stationary and stochastic) policy $\Pi$ from state $s$. Distributions or "lotteries" over these prospects may be represented by (not necessarily unique) tuples of state lottery and policy $(\tilde{s}, \Pi) \in \mathcal{L}(\mathcal{S}) \times \mathbf{\Pi} =: \mathcal{L}(\mathcal{P})$. We write $(\tilde{s}_1, \Pi) \succ (\tilde{s}_2, \Omega)$ if $(\tilde{s}_1, \Pi)$ is strictly preferred to $(\tilde{s}_2, \Omega)$ under preference relation $\succ$.

To be "rational", we require $\succ$ to satisfy the "VNM axioms" [81], which is capture in Axiom 1:

**Axiom 1** (VNM). *For all $\tilde{p}, \tilde{q}, \tilde{r} \in \mathcal{L}(\mathcal{P})$ we have:*

*Asymmetry*: If $\tilde{p} \succ \tilde{q}$, then not $\tilde{q} \succ \tilde{p}$;
*Negative Transitivity*: If not $\tilde{p} \succ \tilde{q}$ and not $\tilde{q} \succ \tilde{r}$, not $\tilde{p} \succ \tilde{r}$;
*Independence*: If $\tilde{p} \succ \tilde{q}$, then $\alpha \tilde{p} + (1 - \alpha)\tilde{r} \succ \alpha \tilde{q} + (1 - \alpha)\tilde{r}, \forall \alpha \in (0, 1]$;
*Continuity*: If $\tilde{p} \succ \tilde{q} \succ \tilde{r}$, then $\exists \alpha, \beta \in (0, 1)$ such that $\alpha \tilde{p} + (1 - \alpha)\tilde{r} \succ \tilde{q} \succ \beta \tilde{p} + (1 - \beta)\tilde{r}$;

*where $\alpha \tilde{p} + (1 - \alpha)\tilde{q}$ denotes the mixture lottery with $\alpha\%$ chance of $\tilde{p}$ and $(1 - \alpha)\%$ chance of $\tilde{q}$.*

Asymmetry and negative transitivity together form the basic requirements of a strict preference relation—equivalent to completeness and transitivity of the corresponding weak preference relation, $\succeq$ (defined as $p \succeq q \Leftrightarrow q \not\succ p$). Independence can be understood as an irrelevance of unrealized alternatives, or consequentialist, axiom: given that the $\alpha\%$ branch of the mixture is realized, preference between $\tilde{p}$ and $\tilde{q}$ is independent of the rest of the mixture (i.e., what could have happened on the $(1 - \alpha)\%$ branch). Finally, continuity is a natural assumption given that probabilities are continuous.

We further require $\succ$ to be *dynamically consistent* [70, 40, 44]:

**Axiom 2** (Dynamic consistency). $(s, a\Pi) \succ (s, a\Omega)$ *if and only if* $(T(s, a), \Pi) \succ (T(s, a), \Omega)$ *where $T(s, a)$ is the distribution over next states after taking action $a$ in state $s$.*

This axiom rules out the irrational behavior in the Procrastinator's Peril, by requiring today's preferences for tomorrow's actions to be the same as tomorrow's preferences. While some of these axioms (particularly independence and dynamic consistency) have been the subject of debate (see, e.g., [44]), note that the standard RL model is *more* restrictive than they require [54].

The axioms produce two key results that we rely on (see Kreps [39] and Pitis [54] for proofs):

**Theorem 1** (Expected utility representation). *The relation $\succ$ defined on the set $\mathcal{L}(\mathcal{P})$ satisfies Axiom 1 if and only if there exists a function $V : \mathcal{P} \to \mathbb{R}$ such that, $\forall \tilde{p}, \tilde{q} \in \mathcal{L}(\mathcal{P})$:*

$$\tilde{p} \succ \tilde{q} \iff \sum_{z \in supp(\tilde{p})} \tilde{p}(z)V(z) > \sum_{z \in supp(\tilde{q})} \tilde{q}(z)V(z).$$

*Another function $V^{\dagger}$ gives this representation iff $V^{\dagger}$ is a positive affine transformation of $V$.*

Using Theorem 1, we extend the domain of value function $V$ to $\mathcal{L}(\mathcal{P})$ as $V(\tilde{p}) = \sum_z \tilde{p}(z)V(z)$.

**Theorem 2** (Generalized Bellman representation). *If $\succ$ satisfies Axioms 1-2 and $V$ is an expected utility representation of $\succ$, there exist $R : S \times A \to \mathbb{R}$, $\gamma : S \times A \to \mathbb{R}^+$ such that $\forall s, a, \Pi$,*

$$V(s, a\Pi) = R(s, a) + \gamma(s, a)V(T(s, a), \Pi).$$

**Remark 3.1.1** Instead of preferences over stochastic processes of potential futures, one could begin with preferences over trajectories [86, 64, 10]. The author takes issue with this approach, however, as it's unclear that such preferences should satisfy *Asymmetry* or *Independence* without additional assumptions (humans often consider counterfactual outcomes when evaluating the desirability of a trajectory) [54]. By using Theorem 2 to unroll prospects, one can extend preferences over prospects to define preferences over trajectories according to their discounted reward.

**Remark 3.1.2** Theorem 2, as it appeared in Pitis [54], required an additional, explicit "Irrelevance of unrealizable actions" axiom, since prospects were defined as tuples $(\tilde{s}, \Pi)$. This property is implicit in our redefinition of prospects as stochastic processes.

**Remark 3.1.3** In this line of reasoning only the preference relation $\succ$ is primitive; $V$ and its Bellman form $(R, \gamma)$ are simply representations of $\succ$ whose existence is guaranteed by the axioms. Not all numerical representations of $\succ$ have these forms [84]. In particular, (strictly) monotonically increasing transforms preserve ordering, so that any increasing transform $V^\dagger$ of a Theorem 1 representation $V$ is itself a valid numerical representation of $\succ$ (although lotteries will no longer be valued by the expectation over their atoms unless the transform is affine, per Theorem 1).

## 3.2 Representing rational aggregation

Let us now consider the aggregation of several preferences. These may be the preferences of an agent's several principals or preferences representing a single individual's competing interests. Note at the outset that it is quite natural for different objectives or principals to have differing time preference. We saw one example in the Procrastinator's Peril, but we can also consider a household robot that seeks to aggregate the preferences of Alice and her husband Bob, for whom there is no reason to assume equal time preference [26].

An intuitively appealing axiom for aggregation is Pareto indifference, which says that if each individual preference is indifferent between two alternatives, then so too is the aggregate preference.

**Axiom 3** (Pareto indifference). *If $\tilde{p} \approx_i \tilde{q}$ ($\forall i \in \mathcal{I}$), $\tilde{p} \approx_\Sigma \tilde{q}$.*

Here, $\tilde{p} \approx \tilde{q}$ means indifference (not $\tilde{p} \succ \tilde{q}$ and not $\tilde{q} \succ \tilde{p}$), so that $\approx_i$ is the $i$th individual indifference relation, $\mathcal{I}$ is a finite index set over the individuals, and $\approx_\Sigma$ indicates the aggregate relation. There exist stronger variants of Pareto property that require monotonic aggregation (e.g., if all individuals prefer $\tilde{p}$, so too does the aggregate; see Axiom 3′ in Subsection 5.1). We opt for Pareto indifference to accommodate potentially deviant individual preferences (e.g., if all individuals are indifferent but for a sociopath, the aggregate preference may be opposite of the sociopath's).

If we require individual and aggregate preferences to satisfy Axiom 1 and, jointly, Axiom 3, we obtain a third key result due to Harsanyi [32]. (See Hammond [31] for proof).

**Theorem 3** (Harsanyi's representation). *Consider individual preference relations $\{\succ_i; i \in \mathcal{I}\}$ and aggregated preference relation $\succ_\Sigma$, each defined on the set $\mathcal{L}(\mathcal{P})$, that individually satisfy Axiom 1 and jointly satisfy Axiom 3. If $\{V_i; i \in \mathcal{I}\}$ and $V_\Sigma$ are expected utility representations of $\{\succ_i; i \in \mathcal{I}\}$ and $\succ_\Sigma$, respectively, then there exist real-valued constant $c$ and weights $\{w_i; i \in \mathcal{I}\}$ such that:*

$$V_\Sigma(\tilde{p}) = c + \sum\nolimits_{i \in \mathcal{I}} w_i V_i(\tilde{p}).$$

*That is, the aggregate value can be expressed as a weighted sum of individual values (plus a constant).*

According to Harsanyi's representation theorem, the aggregated value function is a function of the individual value functions, *and nothing else*. In other words, Pareto indifferent aggregation of VNM preferences that results in VNM preference is necessarily *context-free*—the same weights $\{w_i\}$ apply regardless of state and policy.

We will also make use of two technical conditions to eliminate certain edge cases. Though sometimes left implicit, these are both common requirements for aggregation functions [5].

**Axiom 4** (Technical conditions on $\succ_\Sigma$).

*Unrestricted Domain*: $\succ_\Sigma$ is defined for all valid individual preference sets $\{\succ_i; i \in \mathcal{I}\}$.
*Sensitivity*: $\forall i \in \mathcal{I}$, holding $\succ_j, j \neq i$ constant, there exist $\succ_i^1, \succ_i^2$ resulting in different $\succ_\Sigma$.

The first condition allows us to consider conflicting objectives with different time preference. The second condition implies that the weights $w_i$ in Theorem 3 are nonzero.

**Remark 3.2.1** It is worth clarifying here the relation between Harsanyi's theorem and a related class of aggregation theorems, occasionally cited within the machine learning literature (e.g., [6]), based on axioms originating with Debreu [19]. One instance of this class states that any aggregate preference satisfying six reasonable axioms can be represented in the form: $V_\Sigma(\tilde{p}) = \sum_{i \in \mathcal{I}} m(V_i(\tilde{p}))$, where $m$ is a strictly increasing monotonic function from the family $\{x^p \,|\, 0 < p \leq 1\} \cup \{\log(x) \,|\, p = 0\} \cup \{-x^p \,|\, p < 0\}$ [47]. If we (1) drop the symmetry axiom from this theorem to obtain variable weights $w_i$, and (2) add a monotonicity axiom to Harsanyi's theorem to ensure positive weights $w_i$ [32], then the only difference between the theorems is the presence of monotonic function $m$. But

note that applying $m$ to each $V_i$ or to $V_\Sigma$ individually does not change the preference ranking they represent; it does, however, decide whether $V_i$ and $V_\Sigma$ are VNM representations of $\succ$. Thus, we can understand Harsanyi's theorem as saying: if $V_i, V_\Sigma$ are VNM representations of $\succ$, then $m$ must be linear. Or conversely, if $m$ is non-linear ($p \neq 1$), $V_i$ and $V_\Sigma$ are not VNM representations.

**Remark 3.2.2**  A caveat of Harsanyi's theorem is that it implicitly assumes that all individual preferences, and the aggregate preference, use the same set of agreed upon, "objective" probabilities. This is normatively justifiable if we use the same probability distribution (e.g., that of the aggregating agent) to impose "ideal" preferences $\succ_i$ on each individual, which may differ from their implicit subjective or revealed preferences [62]. As noted by Desai et al. [20], Harsanyi's theorem fails if the preferences being aggregated use subjective probabilities. Note, however, that the outcome of the "bargaining" construction in Desai et al. [20] is socially suboptimal when the aggregrating agent has better information than the principals, which suggests that effort should be made to unify subjective probabilities. We leave exploration of this to future work.

### 3.3  Impossibility result

None of Theorems 1-3 assume all Axioms 1-4. Doing so leads to our key result, as follows.

**Theorem 4** (Impossibility).  *Assume there exist distinct policies, $\Pi, \Omega, \Lambda$, none of which is a mixture (i.e., convex combination) of the other two, and consider the aggregation of arbitrary individual preference relations $\{\succ_i; i \in \mathcal{I}\}$ defined on $\mathcal{L}(\mathcal{P})$ that individually satisfy Axioms 1-2. There does not exist aggregated preference relation $\succ_\Sigma$ satisfying Axioms 1-4.*

*Sketch of Proof.*  The full proof is in Appendix B. Briefly, we consider $|\mathcal{I}| = 2$ and use Axiom 4 (Unrestricted Domain) to construct mixtures of $\Pi$ and $\Omega$ so that each mixture is considered indifferent to $\Lambda$ by one of the individual preference relations. Then, by applying Theorem 2 and Theorem 3 in alternating orders to the difference between the value of the mixture policy and the value of $\Lambda$, and doing some algebra, we arrive at the equations

$$w_1\gamma_1(s,a) = w_1\gamma_\Sigma(s,a) \quad \text{and} \quad w_2\gamma_2(s,a) = w_2\gamma_\Sigma(s,a), \tag{1}$$

from which we conclude that $\gamma_\Sigma(s,a) = \gamma_1(s,a) = \gamma_2(s,a)$. But this contradicts our assumption that individual preferences $\{\succ_i; i \in \mathcal{I}\}$ may be chosen arbitrarily, completing the proof. $\qquad\square$

**Intuition of Proof.**  Under mild conditions, we can find two policies (a $\Pi/\Omega$ mixture, and $\Lambda$) between which individual preference $\succ_1$ is indifferent, but individual preference $\succ_2$ is not. Then for mixtures of this $\Pi/\Omega$ mixture and $\Lambda$, $\succ_1$ remains indifferent, but the strength of $\succ_2$ changes, so that $\succ_\Sigma$ must have the same time preference as $\succ_2$. By an analogous argument, $\succ_\Sigma$ must have the same time preference as $\succ_1$, leading to a violation of Unrestricted Domain.

The closest results from the economics literature consider consumption streams ($S = \mathbb{R}$) [88, 15, 83]. Within reinforcement learning, equal time preference has been assumed, without justification, when merging MDPs [68, 41] and value functions for different tasks [29].

**Remark 3.3.1**  A consequence of Unrestricted Domain, critical to the impossibility result, is that individual preferences may have diverse time preferences (i.e., different discount functions). If discounts are equal, Markovian aggregation is possible.

**Remark 3.3.2**  Per Remark 3.2.2, by applying Harsanyi's theorem, we are implicitly assuming that all preferences are formed using the same "objective" probabilities over prospects; is there a notion of "objective" time preference that should be used? If so, this would resolve the impossibility (once again, requiring that we impose a notion of ideal preference on individuals that differs from their expressed preference). We leave this consideration for future work.

**Remark 3.3.3**  Theorem 4 applies to the standard RL setup, where $\gamma_1, \gamma_2, \gamma_\Sigma$ are constants.

## 4  Escaping Impossibility with Non-Markovian Aggregation

An immediate consequence of Theorem 4 is that any scalarized approach to multi-objective RL [79] is generally insufficient to represent composed preferences. But the implications run deeper: insofar as general tasks consist of several objectives, Theorem 4 pushes Sutton's reward hypothesis to its

limits. To escape impossibility, the Procrastinator's Peril is suggestive: to be productive, repeat play should not be rewarded. And for this to happen, we must keep track of past play, which suggests that **reward must be non-Markovian, even when all relevant objectives are Markovian**. That is, even if we have settled on some non-exhaustive state representation that is "sufficiently" Markovian, an extra aggregation step could render it no longer sufficient.

**Relaxing Markov Preference**   The way in which non-Markovian rewards (or equivalently, non-Markovian utilities) can be used to escape Theorem 4 is quite subtle. Nowhere in the proofs of Theorems 1-4 is the Markov assumption explicitly used. Nor does it obviously appear in any of the Axioms. The Markov assumption *is*, however, invoked in two places. First, to establish history-independent comparability between the basic objects of preference—prospects $(s, \Pi)$—and second, to extend that comparison set to include "prospects" of the form $(T(s, a), \Pi)$. To achieve initial comparability, Pitis [54] applied a "Markov preference" assumption (preferences over prospects are independent of history) together with an "original position" construction that is worth repeating here:

> [I]t is admittedly difficult to express empirical preference over prospects ... an agent only ever chooses between prospects originating in the same state ... [Nevertheless,] we imagine a hypothetical state from which an agent chooses between [lotteries] of prospects, denoted by $\mathcal{L}(\mathcal{P})$. We might think of this choice being made from behind a "veil of ignorance" [58].   (2)

In other words, to allow for comparisons between prospect $(s_1, \Pi)$ and $(s_2, \Pi)$, we prepend some pseudo-state, $s_0$, and compare prospects $(s_0 s_1, \Pi)$ and $(s_0 s_2, \Pi)$. Markov preference then lets us cut off the history, so that our preferences between $(s_1, \Pi)$ and $(s_2, \Pi)$ are cardinal.

The impossibility result suggests, however, that aggregate preference is *not* independent of history, so that construction 2 cannot be applied. Without this construction, there is no reason to require relative differences between $V(s_1, *)$ and $V(s_2, *)$ to be meaningful, or to even think about lotteries/mixtures of the two prospects (as done in Axiom 1). Letting go of this ability to compare prospects starting in different states means that Theorem 1 is applicable only to sets of prospects with matching initial states, unless we shift our definition of "prospect" to include the history; i.e., letting $h$ represent the history, we now compare "historical prospects" with form $(h, \Pi)$.

Though this does not directly change the conclusion of Theorem 2, $T(s, a)$ in $V(T(s, a), \Pi)$ includes a piece of history, $(s, a)$, and can no longer be computed as $\mathbb{E}_{s' \sim T(s,a)} V(s', \Pi)$. Instead, since the agent is not choosing between prospects of form $(s', \Pi)$ but rather (abusing notation) prospects of form $(sas', \Pi)$, the expectation should be computed as $\mathbb{E}_{s' \sim T(s,a)} V(sas', \Pi)$.

The inability to compare prospects starting in different states also changes the conclusion of Theorem 3, which implicitly uses such comparisons to find constant coefficients $w_i$ that apply everywhere in the original $\mathcal{L}(\mathcal{P})$. Relaxing the application of Harsanyi's theorem to not make inter-state comparisons results in weights $w_i(h)$ that are history dependent when aggregating the historical prospects.

## 4.1   Possibility Result

Allowing the use of history dependent coefficients in the aggregation of $V_i(T(s, a), \Pi)$ resolves the impossibility. The following result shows that given some initial state dependent coefficients $w_i(s)$, we can always construct history dependent coefficients $w_i(h)$ that allow for dynamically consistent aggregation satisfying all axioms. In the statement of the theorem, $\mathcal{L}(\mathcal{P}_h)$ is used to denote the set of lotteries over prospects starting with history $h$ of arbitrary but finite length (not to be confused with the set of lotteries over all historical prospects, of which there is only one). Note that if a preference relation satisfies Axioms 1-2 with respect to $\mathcal{L}(\mathcal{P})$, the natural extension to $\mathcal{L}(\mathcal{P}_{hs})$, $(hs, \Pi_1) \succ (hs, \Pi_2) \iff (s, \Pi_1) \succ (s, \Pi_2)$, satisfies Axioms 1-2 with respect to $\mathcal{L}(\mathcal{P}_{hs})$. Here, we are using $hs$ to denote a history terminating in state $s$.

**Theorem 5** (Possibility). *Consider the aggregation of arbitrary individual preference relations $\{\succ_i; i \in \mathcal{I}\}$ defined on $\mathcal{L}(\mathcal{P})$, and consequently $\mathcal{L}(\mathcal{P}_h)$, $\forall h$, that individually satisfy Axioms 1-2. There exists aggregated preference relations $\{\succ_\Sigma^h\}$, defined on $\mathcal{L}(\mathcal{P}_h)$, $\forall h$, that satisfy Axioms 1-4.*

*In particular, **given** $s, a, V_\Sigma, \{V_i\}, \{w_i(s)\}$, **where (A)** each $V_i$ satisfies Axioms 1-2 on $\mathcal{L}(\mathcal{P})$, and $V_\Sigma$ satisfies Axioms 1-2 on $\mathcal{L}(\mathcal{P}_h)$, $\forall h$, **and (B)** $V_\Sigma(s, a\Pi) = \sum_i w_i(s) V_i(s, a\Pi))$, **then**, choosing*

$$w_i(sas') := w_i(sa) := w_i(s)\gamma_i(s, a) \quad \text{for all } i, s' \tag{3}$$

*implies that $V_\Sigma(sas', \Pi) \propto \sum_i w_i(sa)V_i(s', \Pi)$ so that the aggregated preferences $\{\succ_\Sigma^{sas'}\}$ satisfy Axiom 3 on $\mathcal{L}(\mathcal{P}_{sas'})$. Unrolling this result—$w_i(hsas') := w_i(hs)\gamma_i(s,a)$—produces a set of constructive, history dependent weights $\{w_i(h)\}$ such that Axiom 3 is satisfied for all histories $\{h\}$.*

*Sketch of Proof.* The full proof is in Appendix B. Following the proof of Theorem 4, we arrive at

$$w_1(s)\gamma_1(s,a) = w_1(sa)\gamma_\Sigma(s,a) \quad \text{and} \quad w_2(s)\gamma_2(s,a) = w_2(sa)\gamma_\Sigma(s,a), \tag{4}$$

from which we conclude that:

$$\frac{w_2(sa)}{w_1(sa)} = \frac{w_2(s)\gamma_2(s,a)}{w_1(s)\gamma_1(s,a)}. \tag{5}$$

This shows the existence of weights $w_i(sa)$, unique up to a constant scaling factor, for which $V_\Sigma(T(s,a), \Pi) \propto \sum_i w_i(sa)V_i(T(s,a), \Pi)$, that apply regardless of how individual preferences are chosen or aggregated at $s$. Unrolling the result completes the proof. $\square$

From Theorem 5 we obtain a rather elegant result: rational aggregation over time discounts the aggregation weights assigned to each individual value function proportionally to its respective discount factor. In the Procrastinator's Peril, for instance, where we started with $w_\mathtt{p}(s) = w_\mathtt{w}(s) = 1$, at the initial (and only) state $s$, we might define $w_\mathtt{p}(sps) = 0.5$ and $w_\mathtt{w}(sps) = 0.9$. With these non-Markovian aggregation weights and $\gamma_\Sigma(sp) = 1$, you can verify that (1) the irrational procrastination behavior is solved, and (2) the aggregated rewards for work and play are now non-Markovian.

**Remark 4.1 (Important!)** The discount $\gamma_\Sigma$ is left undetermined by Theorem 5. One might determine it several ways: by appealing to construction 2 with respect to historical prospects in order to establish inter-state comparability, by setting it to be the highest individual discount (0.9 in the Procrastinator's Peril), by normalizing the aggregation to weights to sum to 1 at each step, or perhaps by another method. In any case, determining $\gamma_\Sigma$ would also determine the aggregation weights, per equation 4 (and vice versa). We leave the consideration of different methods for setting $\gamma_\Sigma$ and establishing inter-state comparability of $V_\Sigma$ to future work. (NB: *This is a normative question*, which we leave unanswered. While one can make assumptions, as we will for our numerical example in Subsection 5.2, future research should be wary of accepting a solution just because it seems to work.)

## 4.2 A Practical State Space Expansion

The basic approach to dealing with non-Markovian rewards is to expand the state space in such a way that rewards becomes Markovian [27, 14, 2]. However, naively expanding $\mathcal{S}$ to a history of length $H$ could have $O((|\mathcal{S}| + |\mathcal{A}|)^H)$ complexity. Fortunately, the weight update in equation 4 allows us to expand the state using a single parameter per objective. In particular, for history $hsa$ and objective $i$, we append to the state the factors $y_i(hsa) := y_i(h)\gamma_i(s,a)/\gamma_\Sigma(s,a)$, which are defined for every history, and can be accumulated online while executing a trajectory. Then, given a composition with any set of initial weights $\{w_i\}$, we can compute the weights of augmented state $s_{\text{aug}} = (s, y_i(hs))$ as $w_i(s_{\text{aug}}) = y_i(hs) \cdot w_i$. Letting $\mathcal{L}(\mathcal{P}^{(y)})$ be the set of prospects on the augmented state set, we get the following corollary to Theorem 5:

**Corollary 1.** *Consider the aggregation of arbitrary individual preference relations $\{\succ_i; i \in \mathcal{I}\}$ defined on $\mathcal{L}(\mathcal{P})$, and consequently $\mathcal{L}(\mathcal{P}^{(y)})$, that individually satisfy Axioms 1-2. There exists aggregated preference relation $\{\succ_\Sigma\}$, defined on $\mathcal{L}(\mathcal{P}^{(y)})$, that satisfies Axioms 1-4.*

## 5 Discussion and Related Work

### 5.1 A Fundamental Tension in Intertemporal Choice

The state space expansion of Subsection 4.2 allows us to represent the (now Markovian) values of originally non-Markovian policies in a dynamically consistent way. While this allows us to design agents that implement these policies, it doesn't quite solve the intertemporal choice problem.

In particular, it is known that dynamic consistency (in the form of Koopmans' Stationarity [38]), together with certain mild axioms, implies a first period dictatorship: the preferences at time $t = 1$ are decisive for all time [25] (in a sense, this is the very definition of dynamic consistency!). Generally speaking, however, preferences at time $t \neq 1$ are not the same as preferences at time $t = 1$ (this is

what got us into our Procrastinator's Peril to begin with!) and we would like to care about the value at all time steps, not just the first.

A typical approach is to treat each time step as a different generation (decision maker), and then consider different methods of aggregating preferences between the generations [33]. Note that (1) this aggregation assumes intergenerational comparability of utilities (see Remark 4.1), and (2) each generation is expressing their personal preferences about what happens in *all* generations, not just their own. Since this is a single aggregation, it will be dynamically consistent (we can consider the first period dictator as a benevolent third party who represents the aggregate preference instead of their own). A sensible approach might be to assert a stronger version of Axiom 3 that uses preference:

**Axiom 3′** (Strong Pareto Preference) If $\tilde{p} \succeq_i \tilde{q}$ $(\forall i \in \mathcal{I})$, then $\tilde{p} \succeq_\Sigma \tilde{q}$; and if, furthermore, $\exists j \in \mathcal{I}$ such that $\tilde{p} \succ_j \tilde{q}$, then $\tilde{p} \succ_\Sigma \tilde{q}$.

Using Axiom 3′ in place of Axiom 3 for purposes of Theorem 3 gives a representation that assigns strictly positive weights to each generation's utility. Given infinite periods, it follows (e.g., by the Borel-Cantelli Lemma for general measure spaces) that if utilities are bounded, some finite prefix of the infinite stream decides the future and we have a "finite prefix dictatorship", which is not much better than a first period one.

The above discussion presents a strong case *against* using dynamic consistency to determine a long horizon policy. Intuitively, this makes sense: preferences change over time, and our current generation should not be stuck implementing the preferences of our ancestors. One way to do this is by taking time out of the equation, and optimizing the expected individual utility of the stationary state-action distribution, $d_\pi$ (cf. Sutton and Barto [74] (Section 10.4) and Naik et al. [50]):

$$J(\pi) = \sum_s d_\pi(s) V_\pi(s) \tag{6}$$

Unfortunately, as should be clear from the use of a stationary policy $\pi$, this time-neutral approach falls short for our purposes, which suggests the use of a non-Markovian policy. While optimizing equation 6 would find the optimal stationary policy in the Procrastinator's Peril (`work` forever with $V(\tau_2) = 3.0$), it seems clear that we *should* `play` at least once ($V(\tau_3) = 3.2$) as this hurts no one and makes the current decision maker better off—i.e., simply optimizing (6) violates the Pareto principle.

This discussion exemplifies a known tension in intertemporal choice between Pareto optimality and the requirement to treat every generation equally—it is impossible to have both [16, 42]. In a similar spirit to Chichilnisky [16], who has proposed an axiomatic approach requiring both finite prefixes of generations and infinite averages of generations to have a say in the social preference, we will examine a compromise (to the author's knowledge novel) between present and future decision makers in next Subsection.

## 5.2 N-Step Commitment and Historical Discounting

We now consider two solutions to the intertemporal choice problem that deviate just enough from dynamic consistency to overcome finite period dictatorships, while capturing almost all value for each decision maker. In other words, they are "almost" dynamically consistent. We leverage the following observation: due to discounting, the first step decision maker cares very little about the far off future. If we `play` once, then `work` for 30 steps in the Procrastinator's Peril, we are already better off than the best stationary policy, regardless of what happens afterward.

This suggests that, rather than following the current decision maker forever, we allow them commit to a non-Markovian policy for some $N < \infty$ steps that brings them within some $\epsilon$ of their optimal policy, without letting them exert complete control over the far future. To implement this, we could reset the accumulated factors $y_i$ to 1 every $N$ steps. This approach is the same as grouping every consecutive $N$ step window into a single, dynamically consistent generation with a first period dictator.

A more fluid and arguably better approach is to discount the past when making current decisions ("historical discounting"). We can implement historical discounting with factor $\eta \in [0, 1]$ by changing the update rule for factors $y_i$ to

$$y_i(hsa) := \eta \left[ y_i(h) \frac{\gamma_i(s, a)}{\gamma_\Sigma(s, a)} \right] + (1 - \eta) w_i^n(s),$$

where $w_i^n(s)$ denotes the initial weight for objective $i$ at the $n$th generation (if preferences do not change over time, $w_i^n = w_i$). This performs an exponential moving average of preferences over time—

high $\eta < 1$ initially gives the first planner full control, but eventually discounts their preferences until they are negligible. This obtains a qualitatively different effect from $N$-step commitment, since the preferences of all past time steps have positive weight (by contrast, on the $N$th step, $N$-step commitment gives no weight to the $N-1$ past preferences). As a result, in the Procrastinator's Peril, any sufficiently high $\eta$ returns policy $\tau_3$, whereas $N$-step commitment would `play` every $N$ steps.

Historical discounting is an attractive compromise because it is both Pareto efficient, and also anonymous with respect to tail preferences—it both optimizes equation 6 in the limit, *and* plays on the first step. Another attractive quality is that historical discounting allows changes in preference to slowly deviate from dynamic consistency over time—the first period is *not* a dictator.

As a numerical example, we can consider what happens in the Procrastinator's Peril if we start to realize that we value occasional play, and preferences shift away from the `play` MDP to a `playn` MDP. The `playn` MDP has fixed $\gamma_{\texttt{playn}} = 0.9$ and non-Markovian reward $R(\texttt{p} \,|\, \text{no play in last N-1 steps}) = 0.5$, $R(\text{anything else}) = 0$. We assume preferences shift linearly over the first 10 timesteps, from $w_{\texttt{play}} = 1, w_{\texttt{playn}} = 0$ at $t = 0$ to $w_{\texttt{play}} = 0, w_{\texttt{playn}} = 1$ at $t = 10$ (and $w_{\texttt{work}} = 1$ throughout). Then, the optimal trajectories $\tau^*$ for different $\eta$, and resulting discounted reward for the *first* time step, $V^1(\tau^*)$, are as follows:

| $\eta$ | $\tau^*$ | $V^1(\tau^*)$ |
| --- | --- | --- |
| 0.00 | `play` for 5 steps, then `play` every 10 steps | 2.635 |
| 0.30 | `play` for 5 steps, then `play` every 10 steps | 2.635 |
| 0.50 | `play` for 3 steps, then `play` every 10 steps | 2.932 |
| 0.90 | `play`, then `work` for 14 steps, then `play` every 10 steps | 3.105 |
| 0.95 | `play`, then `work` for 23 steps, then `play` every 10 steps | 3.163 |
| 0.98 | `play`, then `work` for 50 steps, then `play` every 10 steps | 3.198 |
| 1.00 | `play`, then `work` forever (same as $\tau_3$) | 3.200 |

When $\eta = 0$, each time step acts independently, and since the `play` MDP has high weight at the start, we experience a brief period of consistent `play` in line with the original Procrastinator's Peril, before preferences fully shift to occasional play. With high $\eta < 1$, we capture almost all value for the first time step, while also eventually transitioning to the equilibrium "`play` every 10 steps" policy.

### 5.3 Extension to Boltzmann Policies

The preference relation $\succ$ is deterministic, but the associated policy does not have to be. In many cases—partially-observed, multi-agent, or even fully-observed settings [74, 30]—stochastic policies outperform deterministic ones. And for generative sequence models such as LLMs [13], stochastic policies are inherent. We can extend our analysis—impossibility, possibility, state space expansion, and intertemporal choice rules—to such cases by adopting a stochastic choice rule.

To formalize this, we will take the stochastic choice rule as primitive, and use it to define a (deterministic) relation $\succ$ that, for practical purposes, satisfies Axioms 1-4 [18]. We assume our choice rule satisfies Luce's Choice Axiom [43], which says that the relative rate of choosing between two alternatives in a choice set of $n$ alternatives is constant regardless of the choice set. This can be implemented numerically with a Bradley-Terry choice model [11] by associating each alternative $a_i$ with a scalar $\Omega(a_i)$, so that given choice set $\{a_i, a_j\}$, $p(a_i) = \Omega(a_i)/(\Omega(a_i) + \Omega(a_j))$. (An interesting interpretation for $\Omega(a_i)$ that connects to both probability matching [82, 65] and statistical mechanics [45] is as the number of "outcomes" (microstates) for which $a_i$ is the best choice.)

We then simply "define" preference as $a_i \succ a_j \iff \Omega(a_i) > \Omega(a_j)$, and utility as $V(a_i) := k \log \Omega(a_i)$, so that the policy is a softmax of the utilities. The way these utilities are used in practice (e.g., [30, 17]) respects Axioms 1-2. And summing utilities is a common approach to composition [21, 29], which is consistent with Harsanyi's representation (Theorem 3). For practical purposes then, the impossibility result applies whenever composed objectives may have different time preference.

**Remark 5.3** Unlike our main results, this extension to Boltzmann policies is motivated by practical, rather than normative, considerations. Simple counterexamples to Luce's Choice Axiom exist [87] and probability matching behavior is evidently irrational in certain circumstances [65]. We note, however, that certain theoretical works tease at the existence of a normative justification for Boltzmann policies [18, 8, 73, 23]; given the practice, a clear justification would be of great value.

### 5.4 Related Work in RL

**Task Definition** Tasks are usually defined as the maximization of expected cumulative reward in an MDP [74, 55, 67]. Preference-based RL [86] avoids rewards, operating directly with preferences (but note that preference aggregation invokes Arrow's impossibility theorem [5, 48]), while other works translate preferences into rewards [17, 72, 12, 35]. This paper joins a growing list of work [54, 1, 76, 77] that challenges the implicit assumption that "MDPs are enough" in many reward learning papers, particularly those that disaggregate trajectory returns into stepwise rewards [22, 56, 59].

**Discounting** Time preference or discounting can be understood as part of the RL task definition. Traditionally, a constant $\gamma$ has been used, although several works have considered other approaches, such as state-action dependent discounting [85, 66] and non-Markovian discounting [24, 63]. Several works have considered discounting as a tool for optimization [80] or regularization [36, 4, 57].

**Task Compositionality** Multi-objective RL [61] represents or optimizes over multiple objectives or general value functions [75], which are often aggregated with a linear scalarization function [7, 3]. Rather than scalarizing to obtain a single solution to a multi-objective problem, one can also seek out sets of solutions, such as the set of Pareto optimal policies [78] or the set of acceptable policies [46]. Several works have also considered formal task decompositions [14, 51] where simple addition of MDPs is insufficient [68]. More broadly, in machine learning, composition can be done via mixtures of experts and/or energy-based modeling [34, 21], which have also been applied to RL [29, 41]. Our results provide normative justification for linear scalarization when time preference is the same for all objectives, but call for non-Markovian adjustments when time preferences differ.

**Non-Markovian Rewards** The necessity of non-Markovian rewards was demonstrated in other settings by Abel et al. [1] and more recently, in a concurrent work by Skalse and Abate [69]. Though several papers explicitly consider RL with non-Markovian rewards [27, 60], this is usually motivated by task compression rather than necessity, and the majority of the RL literature restricts itself to Markovian models. Many popular exploration strategies implicitly use non-Markovian rewards [37, 53]. Our work is unique in that non-Markovian rewards arise from aggregating strictly Markovian quantities, rather non-Markovian quantities present in the task definition or algorithm.

## 6 Conclusion and Future Work

The main contribution of this work is an impossibility result from which one concludes that non-Markovian rewards (or an equivalent state expansion) are likely necessary for agents that pursue multiple objectives or serve multiple principals. It's possible that this will be the case for any advanced agent whose actions impact multiple human stakeholders. To accurately align such agents with diverse human preferences we need to endow them with the capacity to solve problems requiring non-Markovian reward, for which this paper has proposed an efficient state space expansion that uses one new parameter per aggregated objective. While the proposed state space expansion allows multi-objective agents to have dynamically consistent preferences for future prospects, it does not, in itself, solve the intertemporal choice problem. To that end, we have proposed "historical discounting", a novel compromise between dynamic consistency and fair consideration of future generations.

Interesting avenues for future work include quantifying the inefficiency of Markovian representations, investigating normative approaches to aggregating preferences based on subjective world models (Remark 3.2.2), considering the existence of an "objective" time preference (Remark 3.3.2), improving methods for determining subjective time preference (e.g., [63]), implementing and comparing approaches to determining $\gamma_\Sigma$ (Remark 4.1), investigating historical discounting in more detail (Subsection 5.2), and considering the existence of a normative justification for Boltzmann policies and their composition (Subsection 5.3).

## Acknowledgments and Disclosure of Funding

I thank Elliot Creager, for a fruitful discussion that prompted Subsection 5.1 and motivated me to turn this into a full paper; Duncan Bailey, who assisted with an earlier workshop version; the anonymous reviewers, who provided detailed reviews and suggestions that helped improve the final manuscript; and Jimmy Ba and the Ba group for early discussions. This work was supported by an NSERC CGS D Award and a Vector Research Grant.

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

# A   Notation Glossary

Table 1: Notation summary.

**Basic notation**

| | |
|---|---|
| $\mathcal{L}(\cdot)$ | the set of probability distributions over its argument |
| $\{\cdot\}$ | a set of its argument |
| $\mathcal{M}$ | a generalized MDP $(\mathcal{S}, \mathcal{A}, \mathcal{T}, \mathcal{T}_0, R, \gamma)$ |
| $\mathcal{S}, s$ | the state space $\mathcal{S}$ with generic state $s \in \mathcal{S}$ |
| $\mathcal{A}, a$ | the action space $\mathcal{A}$ with generic action $a \in \mathcal{A}$ |
| $\mathcal{T}$ | the transition function $\mathcal{T} : \mathcal{S} \times \mathcal{A} \to \mathcal{L}(\mathcal{S})$ |
| $R$ | the reward function $R : \mathcal{S} \times \mathcal{A} \to \mathbb{R}$ |
| $\gamma$ | the generalized discount factor $\gamma : \mathcal{S} \times \mathcal{A} \to \mathbb{R}^+$ |
| $\tau$ | a trajectory $\tau = [(s_0, a_0), (s_1, a_1), \dots]$, which may or may not terminate |
| $h$ | a history (trajectory to up to time $t$) |
| $\pi, \omega$ | stationary policies $\pi, \omega : \mathcal{S} \to \mathcal{L}(\mathcal{A})$ |
| $\Pi, \Omega, \Lambda$ | non-stationary policies $(\pi_t \,\vert\, \tau_0^t, \pi_{t+1} \,\vert\, \tau_0^{t+1}, \dots)$ |
| $\boldsymbol{\Pi}$ | policy space, so $\pi, \omega, \Pi, \Omega \in \boldsymbol{\Pi}$; note $\mathcal{L}(\boldsymbol{\Pi}) = \boldsymbol{\Pi}$ |
| $\mathcal{P}$ | the space $\mathcal{P} = \mathcal{S} \times \boldsymbol{\Pi}$ of prospects over which preferences are expressed; note $\mathcal{L}(P) = \mathcal{L}(S) \times \boldsymbol{\Pi}$ |
| $\mathcal{P}_h$ | the set of historical prospects starting with $h$; NB: $(h_2, \Pi) \notin \mathcal{P}_{h_1}$ for $h_1 \neq h_2$. |
| $\mathcal{P}^{(y)}$ | the set of prospects on augmented state space $\mathcal{S} \cup \{y_i\}$ |
| $(s, \Pi) \in \mathcal{P}$ | a prospect (stochastic process representing the controlled future) |
| $(h, \Pi)$ | a historical prospect (state space has been expanded to include $h$) |
| $hs, sa, s\Pi, a\Pi, \dots$ | concatenations of subcomponents of trajectories/histories/policies; for example, the non-stationary policy $s\Pi := (a, \Pi_0, \Pi_1, \dots)$ |
| $V$ | value function $V : \mathcal{L}(\mathcal{S}) \times \boldsymbol{\Pi} \to \mathbb{R}$, which is equal to the discounted sum of future rewards: $V(\tilde{s}, \Pi) = \mathbb{E}_{s_0 \sim \tilde{s}, \Pi} \left\{ \sum_{t=0}^{\infty} \left[ \prod_{k=1}^{t} \gamma(s_{t-1}, a_{t-1}) \right] r(s_t, a_t) \right\}$ |
| $\mathrm{supp}(\tilde{p})$ | the support of distribution $\tilde{p}$ |
| $\mathcal{I}$ | index set $\mathcal{I}$ of individuals (assumed finite) |
| $w_i, w_i(s), w_i(h)$ | the $i$th aggregation weight, either a constant, or a function of state or history |
| $y_i(h)$ | the $i$th aggregation factor for history $h$ |
| $d_\pi$ | the stationary state-action distribution for policy $\pi$ |

**Generic modifiers**

| | |
|---|---|
| $\cdot_i$ | index $i$ of a sequence, vector, or collection of agents |
| $\cdot_i^j$ | the slice from $i$ to $j$ of a sequence or vector; e.g., $\tau_t^{t+k}$ is the trajectory slice $[(s_t, a_t), \dots, (s_{t+k}, a_{t+k})]$ |
| $\cdot_\Sigma$ | indicates an aggregated item (e.g., $\succ_\Sigma$ is social/aggregated preference) |
| $\cdot'$ | indicates next timestep when time implicit (e.g., $s, s'$) |
| $\tilde{\cdot}$ | indicates a probability distribution (e.g., $\tilde{s} \in \mathcal{L}(\mathcal{S})$) |
| $\succ^h$ | indicates a preference relation on $\mathcal{P}_h$ |

**Operators**

| | |
|---|---|
| $:= \ (=:)$ | defined as (is the definition of) |
| $\succ$ | strict preference |
| $\succeq$ | weak preference $(p \succeq q \leftrightarrow q \not\succ p)$ |
| $\approx$ indifference $(p \not\succ q$ and $q \not\succ p)$ | |

# B Proofs

**Theorem 4** (Impossibility). *Assume there exist distinct policies, $\Pi, \Omega, \Lambda$, none of which is a mixture (i.e., convex combination) of the other two, and consider the aggregation of arbitrary individual preference relations $\{\succ_i; i \in \mathcal{I}\}$ defined on $\mathcal{L}(\mathcal{P})$ that individually satisfy Axioms 1-2. There does not exist aggregated preference relation $\succ_\Sigma$ satisfying Axioms 1-4.*

*Proof.* Fix $s, a$. Using Theorem 2, choose $\{r_i, \gamma_i\}, r_\Sigma, \gamma_\Sigma$ to represent individual and aggregate preferences over $\mathcal{L}(\mathcal{P})$. Define mixture policy $\Pi_\beta := \beta\Pi + (1 - \beta)\Omega$. W.l.o.g. assume $|\mathcal{I}| = 2$.

We use Axiom 4 (Unrestricted Domain) to set

$$
\begin{aligned}
(T(s,a), \Pi) \succ_1 (T(s,a), \Lambda) \succ_1 (T(s,a), \Omega), \\
(T(s,a), \Omega) \succ_2 (T(s,a), \Lambda) \succ_2 (T(s,a), \Pi),
\end{aligned}
\tag{7}
$$

and, using Theorem 1 to shift $V_1, V_2$, we have

$$
\begin{aligned}
V_1(T(s,a), \Lambda) &= V_2(T(s,a), \Lambda) = 0, \\
V_1(T(s,a), \Pi) &> 0 > V_1(T(s,a), \Omega) \quad \text{s.t.} \quad V_1(T(s,a), \Pi_{\beta_1}) = 0, \\
V_2(T(s,a), \Omega) &> 0 > V_2(T(s,a), \Pi) \quad \text{s.t.} \quad V_2(T(s,a), \Pi_{\beta_2}) = 0,
\end{aligned}
\tag{8}
$$

for some $\beta_1, \beta_2 \in (0, 1)$ with $\beta_1 \neq \beta_2$ (again appealing to Unrestricted Domain). Intuitively, equation 7 requires individual preferences to conflict, and the $\beta_1 \neq \beta_2$ condition requires them to be non-symmetric about $\Lambda$. Equation 8 is merely a convenient choice of numerical representation.

We now apply Theorem 3 followed by Theorem 2 to the expression $V_\Sigma(s, a\Pi_\beta) - V_\Sigma(s, a\Lambda)$. We again invoke Theorem 1 to shift $V_\Sigma$ and eliminate the constant term, so that by Theorem 3 $\exists \{w_i\}$ for which,

$$
\begin{aligned}
V_\Sigma(s, a\Pi_\beta) - V_\Sigma(s, a\Lambda) &= \sum_{i \in \mathcal{I}} w_i V_i(s, a\Pi_\beta) - \sum_{i \in \mathcal{I}} w_i V_i(s, a\Lambda) \\
&= w_1 \gamma_1(s,a) \left[ V_1(T(s,a), \Pi_\beta) - V_1(T(s,a), \Lambda) \right] + \\
&\quad w_2 \gamma_2(s,a) \left[ V_2(T(s,a), \Pi_\beta) - V_2(T(s,a), \Lambda) \right] \\
&= w_1 \gamma_1(s,a) V_1(T(s,a), \Pi_\beta) + w_2 \gamma_2(s,a) V_2(T(s,a), \Pi_\beta).
\end{aligned}
\tag{9}
$$

where the second line applies Theorem 2, with rewards cancelling out.

Alternatively, applying Theorem 2 followed by Theorem 3 to the same expression yields,

$$
\begin{aligned}
V_\Sigma(s, a\Pi_\beta) - V_\Sigma(s, a\Lambda) &= \gamma_\Sigma(s,a) V_\Sigma(T(s,a), \Pi_\beta) - \gamma_\Sigma(s,a) V_\Sigma(T(s,a), \Lambda) \\
&= w_1 \gamma_\Sigma(s,a) \left[ V_1(T(s,a), \Pi_\beta) - V_1(T(s,a), \Lambda) \right] + \\
&\quad w_2 \gamma_\Sigma(s,a) \left[ V_2(T(s,a), \Pi_\beta) - V_2(T(s,a), \Lambda) \right] \\
&= w_1 \gamma_\Sigma(s,a) V_1(T(s,a), \Pi_\beta) + w_2 \gamma_\Sigma(s,a) V_2(T(s,a), \Pi_\beta)
\end{aligned}
\tag{10}
$$

Combining equations 9 and 10, we obtain:

$$
\begin{aligned}
& w_1 \gamma_1(s,a) V_1(T(s,a), \Pi_\beta) + w_2 \gamma_2(s,a) V_2(T(s,a), \Pi_\beta) \\
&= w_1 \gamma_\Sigma(s,a) V_1(T(s,a), \Pi_\beta) + w_2 \gamma_\Sigma(s,a) V_2(T(s,a), \Pi_\beta).
\end{aligned}
\tag{11}
$$

Finally, setting $\beta$ to be $\beta_1$ or $\beta_2$ in equation 11, we obtain the equalities:

$$
\begin{aligned}
w_1 \gamma_1(s,a) V_1(T(s,a), \Pi_{\beta_2}) &= w_1 \gamma_\Sigma(s,a) V_1(T(s,a), \Pi_{\beta_2}), \\
w_2 \gamma_2(s,a) V_2(T(s,a), \Pi_{\beta_1}) &= w_2 \gamma_\Sigma(s,a) V_2(T(s,a), \Pi_{\beta_1}).
\end{aligned}
\tag{12}
$$

The $w_i$ are non-zero (Axiom 4, Sensitivity) and the remaining values are non-zero (else $\beta_1 = \beta_2$), so we conclude that $\gamma_\Sigma(s,a) = \gamma_1(s,a) = \gamma_2(s,a)$. But this contradicts our assumption that individual preferences $\{\succ_i; i \in \mathcal{I}\}$ may be chosen arbitrarily, completing the proof. $\qquad \square$

**Theorem 5** (Possibility). *Consider the aggregation of arbitrary individual preference relations $\{\succ_i; i \in \mathcal{I}\}$ defined on $\mathcal{L}(\mathcal{P})$, and consequently $\mathcal{L}(\mathcal{P}_h)$, $\forall h$, that individually satisfy Axioms 1-2. There exists aggregated preference relations $\{\succ_\Sigma^h\}$, defined on $\mathcal{L}(\mathcal{P}_h)$, $\forall h$, that satisfy Axioms 1-4.*

*In particular, **given** $s, a, V_\Sigma, \{V_i\}, \{w_i(s)\}$, **where (A)** each $V_i$ satisfies Axioms 1-2 on $\mathcal{L}(\mathcal{P})$, and $V_\Sigma$ satisfies Axioms 1-2 on $\mathcal{L}(\mathcal{P}_h), \forall h$, **and (B)** $V_\Sigma(s, a\Pi) = \sum_i w_i(s)V_i(s, a\Pi)$), **then**, choosing*

$$w_i(sas') := w_i(sa) := w_i(s)\gamma_i(s, a) \quad \textit{for all } i, s' \tag{13}$$

*implies that $V_\Sigma(sas', \Pi) \propto \sum_i w_i(sa)V_i(s', \Pi)$ so that the aggregated preferences $\{\succ_\Sigma^{sas'}\}$ satisfy Axiom 3 on $\mathcal{L}(\mathcal{P}_{sas'})$. Unrolling this result—$w_i(hsas') := w_i(hs)\gamma_i(s, a)$—produces a set of constructive, history dependent weights $\{w_i(h)\}$ such that Axiom 3 is satisfied for all histories $\{h\}$.*

*Proof.* We follow the proof of Theorem 4. Fix $s, a$. Using Theorem 2, choose $\{r_i, \gamma_i\}, r_\Sigma, \gamma_\Sigma$ to represent individual and aggregate preferences over $\mathcal{L}(\mathcal{P}_s)$ and $\mathcal{L}(\mathcal{P}_{sas'})$. Define mixture policy $\Pi_\beta := \beta\Pi + (1 - \beta)\Omega$. W.l.o.g. assume $|\mathcal{I}| = 2$.

We use Axiom 4 (Unrestricted Domain) to set

$$\begin{aligned} (T(s, a), \Pi) &\succ_1 (T(s, a), \Lambda) \succ_1 (T(s, a), \Omega), \\ (T(s, a), \Omega) &\succ_2 (T(s, a), \Lambda) \succ_2 (T(s, a), \Pi), \end{aligned} \tag{14}$$

and, using Theorem 1 to shift $V_1, V_2$, we have

$$\begin{aligned} V_1(T(s, a), \Lambda) &= V_2(T(s, a), \Lambda) = 0, \\ V_1(T(s, a), \Pi) &> 0 > V_1(T(s, a), \Omega) \quad \text{s.t.} \quad V_1(T(s, a), \Pi_{\beta_1}) = 0, \\ V_2(T(s, a), \Omega) &> 0 > V_2(T(s, a), \Pi) \quad \text{s.t.} \quad V_2(T(s, a), \Pi_{\beta_2}) = 0, \end{aligned} \tag{15}$$

for some $\beta_1, \beta_2 \in (0, 1)$ with $\beta_1 \neq \beta_2$ (again appealing to Unrestricted Domain).

We now apply Theorem 3 followed by Theorem 2 to the expression $V_\Sigma(s, a\Pi_\beta) - V_\Sigma(s, a\Lambda)$. We again invoke Theorem 1 to shift $V_\Sigma$ and eliminate the constant term, so that by Theorem 3 $\exists \{w_i(s)\}$ for which,

$$\begin{aligned} V_\Sigma(s, a\Pi_\beta) - V_\Sigma(s, a\Lambda) &= \sum_{i \in \mathcal{I}} w_i(s)V_i(s, a\Pi_\beta) - \sum_{i \in \mathcal{I}} w_{i(s)}V_i(s, a\Lambda) \\ &= w_1(s)\gamma_1(s, a)\left[V_1(T(s, a), \Pi_\beta) - V_1(T(s, a), \Lambda)\right] + \\ &\quad w_2(s)\gamma_2(s, a)\left[V_2(T(s, a), \Pi_\beta) - V_2(T(s, a), \Lambda)\right] \\ &= w_1(s)\gamma_1(s, a)V_1(T(s, a), \Pi_\beta) + w_2(s)\gamma_2(s, a)V_2(T(s, a), \Pi_\beta). \end{aligned} \tag{16}$$

where the second line applies Theorem 2, with rewards cancelling out.

It suffices to find one set of satisfactory $w_i(h)$, so we can assume that, given $s, a$, $w_i(sas') := w(sa)$ is the same for all $s'$. This will allow us to factor it out below. Then, applying Theorem 2 followed by Theorem 3 to the same expression yields,

$$\begin{aligned} V_\Sigma(s, a\Pi_\beta) - V_\Sigma(s, a\Lambda) &= \gamma_\Sigma(s, a)V_\Sigma(T(s, a), \Pi_\beta) - \gamma_\Sigma(s, a)V_\Sigma(T(s, a), \Lambda) \\ &= w_1(sa)\gamma_\Sigma(s, a)\left[V_1(T(s, a), \Pi_\beta) - V_1(T(s, a), \Lambda)\right] + \\ &\quad w_2(sa)\gamma_\Sigma(s, a)\left[V_2(T(s, a), \Pi_\beta) - V_2(T(s, a), \Lambda)\right] \\ &= w_1(sa)\gamma_\Sigma(s, a)V_1(T(s, a), \Pi_\beta) + w_2(sa)\gamma_\Sigma(s, a)V_2(T(s, a), \Pi_\beta) \end{aligned} \tag{17}$$

Combining equations 16 and 17, setting $\beta$ to be $\beta_1$ or $\beta_2$ in equation, and rearranging, we obtain the equalities:

$$w_1(s)\gamma_1(s, a) = w_1(sa)\gamma_\Sigma(s, a) \quad \text{and} \quad w_2(s)\gamma_2(s, a) = w_2(sa)\gamma_\Sigma(s, a), \tag{18}$$

from which we conclude that:

$$\frac{w_2(sa)}{w_1(sa)} = \frac{w_2(s)\gamma_2(s, a)}{w_1(s)\gamma_1(s, a)} \tag{19}$$

This shows the existence of weights $w_i(sa)$, unique up to a constant scaling factor, for which $V_\Sigma(T(s, a), \Pi) \propto \sum_i w_i(sa)V_i(T(s, a), \Pi)$, that apply regardless of how individual preferences are chosen or aggregated at $s$. Unrolling the result completes the proof. $\qquad\square$

