# OpenReview forum: "Consistent Aggregation of Objectives with Diverse Time Preferences Requires Non-Markovian Rewards"
_NeurIPS.cc/2023/Conference — NeurIPS 2023 poster_

### Official Review · Reviewer_qGsM · 2023-07-05

**Soundness:** 4 excellent
**Presentation:** 2 fair
**Contribution:** 2 fair
**Rating:** 6
**Confidence:** 4

**Summary:**

The paper considers a general multi-objective sequential decision-making setting where each objective may use a different discount factor. Using an axiomatic approach, the authors prove that under some axioms (vNM axioms + dynamic consistency for the relation on each objective), an aggregated preference relation cannot simultaneously satisfy the vNM axioms, dynamic consistency, Pareto indifference, and some technical conditions. In addition, the authors discuss some ways out to this impossibility result, notably via state augmentation or relaxing dynamic consistency.

**Strengths:**

Impossibility result, although the result is actually not very surprising when different discount factors are allowed

Proposition and discussion about different solutions to this impossibility result

**Weaknesses:**

The presentation and organization of the paper could be improved. Notably:

The results could be presented in a more accessible way to a more general audience. The authors seems to know well the related literature in decision theory and economics, which may not necessarily be the case for the NeurIPS audience. For instance, the second and third paragraphs of Section 5.1 are quite hard to follow for a non-expert.

Section 4.2 should be checked. Some notations (e.g., h_{:-1}, y_i, or y) are not explained properly or are not used in a rigorous way.

It is not clear to me why Section 5 combines a presentation of other solutions to the impossibility theorem and a discussion of related work.

As far as I know, most work in multiobjective reinforcement learning applies an identical discount factor on all the objectives. Since the impossibiity theorem doesn't apply in this case, the results of this paper don't apply to most such work. Therefore, I believe researchers may be mislead by the title of this paper. I suggest the authors to use a more precise one.

**Questions:**

None

**Limitations:**

Not applicable, this is a theoretical paper

---

> ### Author Rebuttal · Authors · 2023-08-10
>
> Thank you for your helpful commentary. Please see the General Response above re: most points.
>
> Otherwise, the reason Section 5 is somewhat of a hybrid is because (1) the parts about intertemporal choice and stochastic preference necessarily introduce "related work" that wouldn't fit, or would be duplicative in a standalone related work section, and (2) having a standalone related work section for just reinforcement learning work seems odd. We're not sure how to improve this, but welcome any suggestions.

---

> > ### Comment · Reviewer_qGsM · 2023-08-15
> > **Response**
> >
> > Thank you for the rebuttal. I believe it has mostly addressed the issues I raised. For now, I will keep my score unchanged.

---

### Official Review · Reviewer_j2th · 2023-07-05

**Soundness:** 4 excellent
**Presentation:** 3 good
**Contribution:** 3 good
**Rating:** 6
**Confidence:** 4

**Summary:**

The authors analyze the implications of preference aggregation within a Markov Decision Process Framework. They show that it is not possible to ensure dynamic consistency in an aggregated MDP if one also wants to be able to accommodate arbitrary preference criteria, even if the criteria are individually dynamically consistent. They further show that by relaxing the Markov condition incrementally, dynamic consistency can be recovered.


**Strengths:**

Addresses a fundamental representational issue in MDPs. The authors clearly have a deep understanding of the temporal consistency literature in economics and decision theory, and bring it to bear here. The technical exposition is clear, and the reasoning is sound. The example of procrastination is instructive. The construction of a patch to deal with different discount rates is perhaps the most directly useful contribution. There is also some interesting extended discussion about intemporal preferences that could be quite relevant in an AI context.


**Weaknesses:**

The authors attempt to motivate the contribution in the context of the current "Reward is Enough" debate in RL. This is tantalizing, but seems to me a bit forced. Really we have a basic technical question in representation of intertemporal preferences, and the paper underlines a common lesson that aggregating across separate preferences is never as straightforward as one might think. The technical points are perhaps connected to some arguments debate participants have brought up (and they should indeed be better versed in intertemporal choice), but ultimately we face the same questions we always do about how much and what kind of state should we incorporate to keep things approximately enough Markovian.


**Questions:**

Technical Comments

C1. The whole concept of a discount factor in MDPs is to represent a global time preference. Why should anyone have thought it could be coherent to have different time preferences on different criteria? To the extent that discounting is really capturing dynamics in the world (e.g., consequences of work persisting), then arguably that would more properly be represented in the state space to begin with. That is, using the discount factor for this is a hack and one should not be surprised it is fragile.

C2. Axiom 3 is really strong. It basically entails strong independence of criteria--what is needed to get an additive representation. Many readers will not realize that. Moreover, I suspect it is much stronger than needed to get your key impossibility result. That is, even forms with many more interaction terms will probably run into problems with temporal consistency.

C3. On reflection, it seems to me that the key technical point here could be restated as saying that given Axiom 3, the *only* thing that can go wrong is mismatch of discount factors. Do you agree with that characterization?

Quibbles

Q1. About the procrastination story. Line 67 says that the policy must remember it had previously chosen "play" in order to work forever. Not true (it actually does not matter what was executed in the first step): it just needs to know it is beyond the first step.

Q2. Title. "Multi-objective agency" is a very ill-defined term, so hard to buy the assertion it requires anything in particular. What is actually proved here is about what is required for dynamically consistent preference aggregation.

Q3. Exposition had a few minor gaps, for example \cal{P} never formally defined (clear from context, though).

**Limitations:**


The paper adequately discusses assumptions and limitations.

---

> ### Author Rebuttal · Authors · 2023-08-10
>
> Thank you for your detailed review and helpful commentary/questions.
>
> > The authors attempt to motivate the contribution in the context of the current "Reward is Enough" debate in RL. This is tantalizing, but seems to me a bit forced. …
>
> We think it is relevant for the following reason:
>
> - Suppose we have decided “how much and what kind of state” keeps things “approximately enough Markovian” for Alice and Bob, who have different preferences. Now suppose Alice and Bob purchase an LLM-based personal assistant Carl, who inherits (i.e. aggregates) their preferences. *The same state that was sufficiently Markovian for Alice and Bob may not be sufficiently Markovian for Carl.*
>
> If Carl now becomes a principal (e.g. it delegates some work to household robot Dave), we may need to again adjust our definition of "sufficiently Markovian" for Dave. And so on.
>
> ---
>
> **C1:** Please see general response. To the extent that different humans can have different time preferences, then agents serving multiple humans will face this problem. We think discounting captures more than just the “dynamics in the world” (and can actually be considered entirely apart from dynamics, as is the case in work that analyzes discounting of consumption streams (e.g. Koopmans 1960/Diamond 1965, etc.)). From a representational standpoint, we agree that the state space can be used to absorb everything, but such a representation may give up the compression advantages of Markovian rewards/discounts.
>
> **C2:** “I suspect it is much stronger than needed to get your key impossibility result.” <- this is a really interesting idea, which we will consider (at least for future work). That being said, while the consequence of Axiom 3 (together with VNM) is strong, we do not think the Axiom itself is strong: we think it is difficult to come up with a reasonable case where society (the agent) should prefer A to B if all members of society (the principals) are indifferent between A and B. Variations of the pareto axiom are commonly assumed to be desirable (e.g. for Arrow’s Impossibility Theorem and other work on social welfare functions, and the works cited in Subsection 5.1).
>
> **C3:** Yes. Mismatch of discount factors is the only way Theorem 4 takes effect. That said, per our general response, we argue that mismatch of time preference is the general case whenever there are multiple principals.
>
> **Q1:** Good point, thank you. This could definitely be repaired by a more complex example, but TBD if we can maintain the simplicity while keeping the original claim. We will either correct the claim or tweak the example slightly.
>
> **Q2-Q3:** See general response.

---

> ### Comment · Reviewer_j2th · 2023-08-11
>
> I have read the author rebuttal. I particularly appreciate the authors' willingness to change the title.
> My overall evaluation of the paper is unchanged.
>
> A lot of the response appeals to an interpretation of criteria as multiple agents. Under that interpretation, we should be even less surprised that aggregation breaks things. Not just the Markov structure, but even the possibility of having a single reward function that maintains desirable properties of the collective.

---

### Official Review · Reviewer_WDhN · 2023-07-30

**Soundness:** 4 excellent
**Presentation:** 2 fair
**Contribution:** 3 good
**Rating:** 6
**Confidence:** 3

**Summary:**

This paper examines multi-objective reinforcement learning---the setting in which multiple distinct objectives are desired, and often combined, to form a composite objective. Concretely, the paper explores the limits of aggregating different objectives by appealing to three main pools of ideas. First, to the von Neumann-Morgenstern expected utility, axioms; Second, to Pareto indifference; and Third, to dynamic consistency. The main result is an impossibility result, illustrating that objectives with different time-based objectives (that is, different discount factors), cannot be aggregated in a way that yields a Markovian reward function even if the individual objectives themselves are Markovian. The paper then explores what lies beyond this impossibility result, exploring a mechanism for expanding the state-space to collapse the non-Markovian aggregated objective down to a Markovian one. This results in a "historical" discount factor, a hindsight view of the discount factor.

**Strengths:**

**STRENGTHS**

The paper possesses many strengths:
1) The aspirations of the work are ambitious and important. Clearly establishing when certain kinds of objectives can and cannot be captured is important.
2) The work is rigorous, and well-connected to classical results in decision theory.
3) The examples are clear and help to communicate the main ideas.
4) The impossibility result on its own is interesting. Once I understood the details, it is not ultimately surprising, but I do not believe the result needs to be surprising to be useful.
5) Historical discounting is a new and interesting idea.


**Weaknesses:**


**WEAKNESSES**

At the same time, the paper has several weaknesses:
1) Language. First, and my biggest critique, the work commonly makes use of unusual and vague language surrounding some of the main concepts. For instance, the title, and central idea of the work---multi-objective agency---is not well-defined, and by my reading is not an appropriate choice of description for the content of the work. I would recommend moving away from "agency" as a term, and certainly "multi-objective agency", as neither are well defined in this paper. Instead, I would suggest using "multi-critieria objectives", or "multi-objective RL", as is used in prior literature. It is much more clear and precise, and more well connected with the work.
2) Clarity, and detail of exposition. Ideas are often introduced abruptly and not explained in much detail. For instance, the axioms in section 3.1 are simply stated without any added context or explanation. While vNM is quite common, dynamic consistency is less so, and likely deserves a more thorough, careful, and simple explanation, given its central role in the work. Similarly, after some theorems are introduced, they are sometimes not discussed. Other details are often left unexplained, such as "...none of which is a mixture of the other two" in Theorem 4 (the main result). From a quick reading it is possible to understand this, but it would be worthwhile to spell this out carefully.
3) Notation is sometimes defined quite precisely, but it is often overly complex or not defined. For instance, in Theorem 5, it is unclear what {$\succ_\Sigma^{sas'}$} is intended to mean. Conventions tend to deviate quite a lot from typical work in reinforcement learning as well (getting rid of the Q function in favor of two uses of V, using $\Pi$ instead of $\pi$).
4) Unsurprising results. Lastly, I do believe most of the results are unsurprising. When discount factors vary across objectives, it is perhaps expected that their aggregation will not be representable in the same form (and that we can augment the state space to remedy this). Still, it  is useful to make these arguments carefully and rigorously.

**Questions:**

Primary recommendation: My strongest recommendation is to move Section 4 to the Appendix, and to use the remaining space to provide additional clarity and exposition around key ideas around results. I believe all of the Axioms need more careful and simpler introductions and discussions (and especially Axiom 3), as well as the main results. I believe this switch will strengthen the paper considerably.

Main Comments/Questions:
- "Our main contribution is an impossibility result from which one concludes that non-Markovian rewards are likely necessary for RL agents that pursue multiple objectives or serve multiple principals". This seems slightly too strong, by my reading of the main result---it really only applies when the discounts across the objectives differ. Is this the role that "likely" is playing in this statement? If so I might suggest adjusting the language to be more precise.
- Separately, I wonder if the paper can comment on when the discount should be associated with an _agent_ rather than an _objective_ in isolation.
- Two pieces of related work come to mind. First, the expressivity of "multi-dimensional" reward was explored by Miura in 2022: On the Expressivity of Multidimensional Markov Reward at the RLDM workshop on RL as Agency. I do wonder about the connections between these two sets of results. Second, Tasse et al. propose an algebra on tasks in RL in "A boolean task algebra for reinforcement learning". I wonder about how this style of composition bears on the findings and perspectives from the present paper.

Small Questions:
- "While this allows us to design agents that implement these policies, it doesn’t quite solve the intertemporal choice problem": what is meant by this statement? Actually designing agents that implement these policies does not seem to be in the purview of this paper. Instead, we can aggregate objectives in a way that allows us to _incentivize_ agents according to an appropriate objective, but this is different from the agent design process.
- I do not understand Axiom 3 as stated and the explanation of the notation below is too brief to elucidate. Given the importance of the Axiom, I encourage expanding the explanation.

Typos and writing suggestions:
- As mentioned above, I think the phrase "multi-objective agency" is not the appropriate way to describe the main concept of the paper. I suggest replacing "multi-objective agency" with "multi-criteria objectives", or "multiple objectives", or some variant thereof.
- I don't believe "(history dependent)" is needed in the abstract following "non-Markovian"
- It looks as thought reward functions are defined as both $R$ (in the definition of the MDP) and $r$ (in, say, Theorem 2). I would encourage picking one use throughout.

---

> ### Author Rebuttal · Authors · 2023-08-10
>
> Thank you for your detailed review and actionable suggestions. Please see the general response for the most important points. Otherwise:
>
> **Primary Rec:** Regarding your suggestion to move Section 4 to the Appendix, we assume you are referring to just the “Relaxing Markov Preference” section (up to Subsection 4.1). We think perhaps Subsection 5.3 is an alternative candidate for the Appendix. In any case, we will do our best to improve the exposition per your suggestion.
>
> **Comment 1:** That was indeed the role that “likely” was meant to play (see ** in Global discount factor in General Response); we will adjust.
>
> **Comment 2:** Our bias is toward viewing discount factors in the economic sense, as representations of “time preference”, which arises naturally when considering preferences over temporal processes / trajectories / streams of consumption. This view would make the discount factor a property of “preferences”, which could either be expressed globally, as the preferences of a complex agent (e.g. a household robot), or individually with respect to a particular objective (e.g. the paperclip making agent). We are not sure if this is responsive to your comment (we see agent and objective as being entangled), but it is in contrast to a rather common view in RL that discounting is either a mathematical convenience, or done for purposes of optimization or regularization. We will consider adding some commentary on this in Subsection 5.4 (Related work in RL).
>
> **Comment 3:**
>
> - *Miura 2023*: Thank you, we were not aware of this work. Based on our read, Miura shows that any "set of acceptable policies" (as previously defined in Abel et al. 2021) can be represented as the solution set to a constraint-based multi-objective problem (i.e., all acceptable policies, and only acceptable policies, perform better than some lower bound with respect to all objectives). This type of constraint-based composition therefore reduces to a *set* of (equally preferred) policies, whereas the scalarization-based composition in our work reduces to a single social reward function (preference ranking). The fact that Markovian scalarization is insufficient to represent a "set of acceptable policies" was shown in Abel et al. 2021.
>
> - *Tasse et al. 2020*: In the Boolean Task Algebra, Tasse et al. consider a particular family of "goal achievement" tasks that make it possible to precisely compose tasks using boolean operations (min/max). Like our work, it is a scalarization-based approach to value function composition. Although it appears as non-linear scalarization (via min/max), we think it can be understood as taking the extreme of the linearly scalarized soft-value function composition approach taken by Haarnoja et al. 2018 and Van Niekerk et al 2019, which makes it related to the discussion in our Subsection 5.3. Unlike our work, it is restricted to a particular family of goal achievement tasks; furthermore they assume their tasks are terminating/undiscounted, so our results would not apply.
>
> **Question 1:** You are correct, thank you. We will adjust.
>
> **Question 2:** As applied to voting / social choice, Axiom 3 says that if all members in a society (indexed by $\mathcal{I}$) are indifferent between two (lotteries of) alternatives ($\tilde p$ and $\tilde q$), then so too is society. We will improve the exposition here.

---

> > ### Comment · Reviewer_WDhN · 2023-08-17
> > **Response to Rebuttal**
> >
> > I thank the authors for their thorough response to each of the reviews. The authors comments and responses to my questions have helped.
> >
> > A few follow up comments:
> > - On the "primary recommendation", I did intend to suggest to move all of Section 4 to the appendix, and to remove emphasis on the point about escaping impossibility from the paper. It is of course up to you, but my suggestion would be to focus purely on the main idea of the paper, which is the impossibility surrounding aggregating non-Markov objectives with different time-preferences. This would give you more time and space to really flesh out the intuition of some of the axioms and the main result.
> > - Comments 1 and 2: that makes sense, thanks.
> > - Comment 3: Thanks, that helps. I believe some commentary about the differences to these two works could strengthen the paper, simply because they are other known ways of composing tasks that do work (at least in the case of the task algebra).

---

### Author Rebuttal · Authors · 2023-08-09

We thank the reviewers for their time and detailed reviews. We find the reviewers have understood our work and have provided helpful suggestions. We are acting on several of the suggestions, as noted below, and agree the changes will improve the paper. We welcome any additional feedback and questions during the discussion period.

General responses / planned changes:

- **Title / “Multi-Objective Agency” (WDhN, j2th, qGsM):** Upon reflection we agree this was not a good choice, and will change the title. We are considering alternatives that move away from the word “agency” and are explicit about “diverse time preference”, along the lines of:
    - Aggregating Objectives with Diverse Time Horizons Requires Non-Markovian Rewards
    - Non-Markovian Aggregation of Objectives with Diverse Time Horizons
    - Where Markovian Scalarization Fails: Aggregating Objectives with Diverse Time Preferences

	We will make similar tweaks throughout to improve precision.

- **Global discount factor vs per objective discounting (WDhN, j2th in C1):** If an “Actor” possesses a single global discount function, and there are two such Actors (e.g. Mom and Dad), jointly acting as principals for a third Actor that we will call the “Agent” (e.g. a household Robot), then the Agent has inherited objectives/rewards from its principals that may** have conflicting time preference.

    (**actually, we believe this to be the general case, not merely a rare occurrence; see, e.g., Frederick, Loewenstein, and O’Donoghue. 2002, Section 8 Paragraph 2, “Thus, there is no reason to expect that discount rates should be consistent across different choices”).

- **Unsurprising main result (WDhN, qGsM):** A number of recent works have recognized or used (Markovian) transition dependent discounting (e.g. Bowling et al. 2023), and the initial question that led to this work was actually, “*How might such a transition dependent discount factor naturally arise?*”.

    Our initial hypothesis was that it might arise through aggregation of objectives with different fixed-discount factors (i.e., our initial hypothesis was a possibility). So while we agree that the impossibility of linearly aggregating MDPs with different discounts into an MDP with a fixed discount and the same state space is not surprising (as recognized by, e.g., Singh & Cohn 1998), and we understand why others might view our result as unsurprising, we were, in fact, surprised by the impossibility.

- **Exposition (WDhN, qGsM):** Per your suggestion, we will do our best to make room for some additional exposition around the Axioms/Theorems to make it more accessible to a NeurIPS audience, while maintaining the current content and structure. This may involve moving certain bits to the Appendix, per Reviewer WDhN’s suggestion.

- **Notation (WDhN, j2th, qGsM):** We will double check usage for precision / rigor / consistency and add a notation glossary to the Appendix.

---

### Decision · Program_Chairs · 2023-09-21

**Decision:**

Accept (poster)

**Comment:**

The reviewers unanimously praised the paper for its soundness, and agreed that the core technical contributions will be of interest to the community, and thus I am recommending this paper be accepted. Reviewers further agreed that the title, and one of the primary terms of the work ("Multi-objective agency"), is ill-defined, and best changed to something less contentious. I agree with this recommendation, and from the discussion it sounds as though the authors are willing to make this change. Of the proposed titles, I am most inclined towards "Aggregating Objectives with Diverse Time Horizons Requires Non-Markovian Rewards", but I leave this to the authors' discretion. I otherwise encourage the authors to take the reviewers' other suggestions into account.